biocomplexity/biophysics/applied mathematics

self-propelled particles, flocking, self-organization, collective dynamics, collective motion, swarming

**Author for correspondence:**
Arthur E. B. T. King
e-mail: arthurebtking@gmail.com

# Non-local interactions in collective motion

Arthur E. B. T. King[1,2] and Matthew S. Turner[2,3,4]

[1]Department of Mathematics, [2]Centre for Complexity Science, and [3]Department of Physics, University of Warwick, Coventry CV4 7AL, UK
[4]Department of Chemical Engineering, Kyoto University, Kyoto 615-8510, Japan

AEBTK, 0000-0003-0958-691X; MST, 0000-0002-3441-678X

The collective motion of animal groups often exhibits velocity–velocity correlations between nearest neighbours, with the strongest velocity correlations observed at the shortest inter-animal spacings. This may have been a motivational factor in the development of models based primarily on short-ranged interactions. Here we ask whether such observations necessarily mean that the interactions are short-ranged. We develop a minimal model of collective motion capable of supporting interactions of arbitrary range and show that it represents a counterexample: the strongest velocity correlations emerge at the shortest distances, even when the interactions are explicitly non-local.

## 1. Introduction

Collective motion is ubiquitous and can be observed at many length scales, from active liquid crystals [1] and synthetic colloids [2], to unicellular microswimmers [3] and multicellular organisms such as insects [4], fish [5], birds [6] and humans [7]. A remarkable feature of such contrasting systems is the spontaneous emergence of collective order. Central to understanding such emergent behaviour is the study of information propagation within collectives. Decisions concerning heading direction must spread swiftly enough to ensure group cohesion [8]. Observed emergent behaviour can be understood by discerning how the microscopic interactions between conspecifics lead to macroscopic behaviour of the collective. Empirical studies of particular animal species have revealed a variety of interaction mechanisms, for example: persistent hierarchical interaction networks in pigeon flocks [9], pairwise startle reactions in fish schools [10], zonal interactions in surf scooters [11], dynamic speeding and turning responses in groups of golden shiners [12], short-range forces in locust swarms [13] and scale-free correlations in starling murmurations [14]. A common feature observed in biological collectives is the strong correlation of nearby neighbours' velocities, for example: bird flocks [6], insect swarms [15] and pedestrian crowds [16].

Such fascinating, diverse and complex behaviour in biological collectives has prompted a statistical physics approach to studying collective motion as a phenomenon. Statistical physics models of collective motion, pioneered by Vicsek *et al.* [17], were developed to address whether there exist global features in the behaviour of large groups of collectively moving organisms. Models of this type are sometimes treated as a transport-related, non-equilibrium analogue of ferromagnetic models, with the crucial difference that agents may move off lattice, allowing the onset of long-range order at finite noise, a feature observed in simulation [17,18], and explained by a continuum theory of flocking by Toner & Tu [19]. Such variants of spin models, relying purely on local interactions, have reproduced several qualitative observables in flocks (e.g. [11,20,21]). Numerical models indicate collective motion can arise from local interactions [14], and it has been suggested that global order probably emerges as a result of local interactions between individuals [22,23]. This may explain why there has been a preference for focusing on local interactions when model fitting to biological collectives.

Interaction mechanisms beyond those that are strictly local have been analysed in several studies. In general, these fall into two different categories. The first category of models include a single interaction mechanism that allows both local and non-local interactions, for example limited-range topological models [24] in which alignment, attraction and repulsion are incorporated in a zonal fashion, or vision models [25] which incorporate interactions at all ranges. The second category of models include distinct local and non-local mechanisms, for example local alignment and vision interactions [26], local alignment and global attraction forces [27], local and non-local alignment [28] and stochastic, asynchronous distant-dependent interactions in which agents repel at short distances, attract at long distances and align at intermediate distances [29]. The effect of *purely* non-local interactions on collectives has been less well explored.

The possible role of non-local interactions should be properly considered when the physiological characteristics of the sensory systems of the animals justify it. Not only are long-ranged correlations observed in naturally occurring swarms [30], but many flocking animals rely primarily on vision, which can easily extend well beyond the flock size. A study by Strandburg-Peshkin *et al.* [25] found that collective motion that included interactions based on line-of-sight better reflected the information employed by biological organisms when determining their movement directions. Models that incorporate purely local interactions must ensure that they are able to reproduce the scale invariance of long-ranged correlations among velocity fluctuations. We note that models with non-local interactions can trivially produce long-ranged velocity correlations.

Here we examine the effect of interaction range—explicitly the effect of a non-local interaction—on collective motion. The model presented is a minimal representation of collective motion and demonstrates a principle general to many biological collectives: local correlations in velocity can arise purely from non-local interactions. The biological relevance of this result is that a strong positive correlation between nearby neighbours' velocities is a common observable, for example bird flocks [6], insect swarms [15] and pedestrian crowds [16]. Hence our work demonstrates the potential importance of non-local interactions in generating alignment in biological collectives.

## 2. Model

We employ a modified version of the Vicsek model [17] in which $N$ self-propelled agents ($i = 1, 2, \ldots, N$) move in a square periodic domain of size $L \times L$, rescaled to maintain constant density $\rho = N/L^2 = 4$ throughout. At discrete time $t$ the agents have positions $x_i^t$ and velocities $v_i^t = v_0 \begin{pmatrix} \cos \theta_i^t \\ \sin \theta_i^t \end{pmatrix}$, involving the orientation $\theta_i^t$ and a fixed speed, here $v_0 = 0.3$. We define $\mathcal{S}_i^t(\alpha, n)$ to be the set of agents that the $i$th agent interacts with at time $t$. This set contains $n$ members and is constructed from a distance-ordered list of the other agents, each at successively increasing distances from agent $i$. The closest member included in $\mathcal{S}_i^t$ is the $\alpha$th most distant agent and the furthest is the $(\alpha + n)$th most distant (figure 1).

The orientation is updated according to

$$\theta_i^{t+1} = \arctan\left(\frac{\langle \sin \theta_j^t \rangle_{j \in \mathcal{S}_i^t(\alpha, n)}}{\langle \cos \theta_j^t \rangle_{j \in \mathcal{S}_i^t(\alpha, n)}}\right) + \sigma_i^t, \tag{2.1}$$

where angled brackets denote the average over agents that are members of $\mathcal{S}_i^t(\alpha, n)$. The rotational noise $\sigma_i^t$ is a random variable, redrawn uniformly from the interval $[-\eta/2, \eta/2]$ for each agent at each time-step. The model proceeds in discrete time-steps $t$ with spacing $\Delta t = 1$. Each agent's orientation, hence

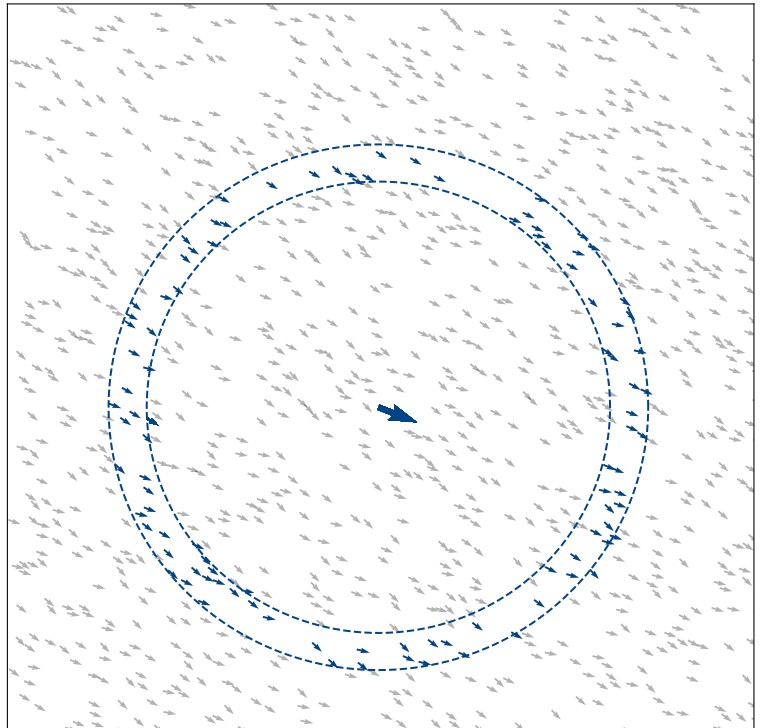

**Figure 1.** A snapshot of agents shown in grey at time $t$, with an arbitrary agent in blue (enlarged for clarity) and its corresponding interacting group (also in blue). In general, each agent co-aligns with all agents in its interacting group. We define the interacting group $\mathcal{S}_i^t(\alpha, n)$ as an ordered list containing $n$ members, each at successive distances from agent $i$, with the closest member being the $\alpha$th most distant. For every agent, the interacting group is recomputed each time-step $t$. The dashed blue circles simply act as a guide to the eye, highlighting the interacting group: their radii are not directly encoded within the model and hence also vary over time. Here $N = 1000$ agents shown with interacting group $\mathcal{S}_i^t(\alpha = 301, n = 100)$.

velocity and position, are updated simultaneously, the latter to

$$x_i^{t+1} = x_i^t + v_i^{t+1}\Delta t. \qquad (2.2)$$

The system undergoes a familiar transition on increasing noise (see electronic supplementary material, figure S1), with order parameter $\Phi = 1/N \langle |\sum_{i=1}^{N} v_i^t/v_0| \rangle_t$ increasing sharply below a crossover noise $\eta_c$. This value is determined for each set of parameters by running an array of simulations to establish the noise value corresponding to the inflection in the order-noise relationship. A line is fit to the noise curve using spline interpolation (univariate one-dimensional smoothing spline of degree $k = 4$) and the crossover point is determined by finding roots of the curve's second derivative using the Newton–Raphson method. Results from this process are listed in electronic supplementary material, table S1. To fairly compare our simulations across different $\alpha$ and $n$ values, we set the noise to $\eta_c$ (unless noted otherwise). This is because different parameter values can otherwise leave one system in the ordered regime and another in the disordered regime. This would make it more difficult to draw comparative conclusions between them, as correlations in the ordered regime would always be *quantitatively* higher. Except perhaps very deep into the disordered regime (where all velocity correlations are anyway lost), the level of noise does not affect our primary qualitative conclusion: that local correlations are strongest even when the co-aligning interactions are between distant agents.

## 3. Observables

We study correlations between agents' velocities in the centre-of-mass frame by defining the relative velocity $u_i^t = v_i^t - 1/N \sum_{k=1}^{N} v_k^t$ and a correlation function

$$C(r) = \left\langle \frac{\sum_{i \neq j} u_i^t \cdot u_j^t \delta(r - r_{ij}^t)}{\sum_{i \neq j} \delta(r - r_{ij}^t)} \right\rangle_t. \qquad (3.1)$$

Here $r_{ij}^t = |\boldsymbol{x}_i^t - \boldsymbol{x}_j^t|$ is the scalar distance between agents $i$ and $j$, and the Dirac delta function selects pairs that are at a mutual metric distance $r$ at time $t$. For numerical purposes, this is mapped to a sequence of finite intervals, or 'bins', for the separations. The angled brackets indicate an average over all time-steps. The normalized correlation function is $\tilde{C}(r) = C(r)/K$ with $K = 1/N \sum_{i=1}^{N} \langle \boldsymbol{u}_i^t \cdot \boldsymbol{u}_i^t \rangle_t$, $K \in [0, v_0^2]$.

We observe that a fixed density $\rho = 4$ corresponds to a typical distance of approximately 0.24 between agents.

All simulations are initialized with agents having uniformly random positions and orientations. All simulations are run for $10^5$ time-steps. The first $10^4$ time-steps are discarded to eliminate initial transients.

# 4. Results

We find the highest velocity correlations are robustly associated with the shortest ranges (nearest neighbours), irrespective of the range of the interactions $\alpha$, see figure 2 (and electronic supplementary material, figures S2–S5 for other values of $N$). A primary peak in the correlation function at the shortest distances is observed in all parameter regimes, including those that involve no explicit interactions between nearest neighbours. The persistence of this primary peak reveals that nearest neighbours have correlated velocities, even when the alignment interaction is non-local $\alpha \sim N$. We can attribute the primary peak to implicit, three-body interactions between agents. Nearest neighbours share similar positions and they will therefore have agents in common in their interacting groups, with which they both co-align (figure 3). This overlap between the interacting groups of two nearby agents leads to an implicit co-alignment between them that is of a three-body character; this relies on agents having a similar perception of agents around them (figure 4).

More generally, these agents can be said to perceive (respond to) a similar sensory environment as a direct result of their physical proximity. While this principle is encoded in our model in a particular manner, the general statement that, 'nearby agents share a similar perception of the distant world' is familiar to us from everyday life. It is natural to speculate that many other, quite different, behavioural models that trace this similarity in perception to a similarity in behaviour might preserve the qualitative features that we report here. These might include interactions that are more natural, given the sensory perception of the corresponding animals and many of these may not exhibit the secondary peak that we observe here—probably an artefact of the sharply delineated interacting groups in our model.

This secondary peak in the correlation functions shown in figure 2 occurs at a range corresponding to the average metric distance $d$ between an agent and members of its interacting group. Figure 2 shows the effect of the interacting group size $n$ on the correlation function. The primary peak grows relative to the secondary peak with increasing $n$. Furthermore, the height of the secondary peak drops with increasing $n$. This result indicates that the strength of implicit, three-body co-alignment grows relative to that of explicit alignment with the interacting group. This could be due to the fact that the primary agent is providing a decreasingly significant contribution to the interacting group of those agents with which it co-aligns, while nearby agents share a substantial fraction of their interacting sets, which increases with the size of $n$.

A metric distance of approximately $2d$ separates a primary agent and members of the interacting groups of members of its own interacting group. A smaller tertiary peak can be observed at $2d$ when this distance is smaller than the largest allowed inter-particle spacing by the box geometry $2d < \sqrt{L}/2$ .

Figure 5 shows that the peak in the correlation function at the shortest inter-particle spacings persists even when the system size is large. This is our main result and we reiterate: there are no explicit interactions at the range associated with this correlation.

Very deep into the disordered regime all velocity correlations are lost (see electronic supplementary material, figure S6). For noise levels both well below and above the crossover noise level (corresponding to more ordered and disordered states respectively), we observe the primary correlation peak at short ranges is preserved (see figure 6).

## 4.1. Heterogeneity in the range between interacting groups

We seek to test our interpretation that nearby agents have correlated velocities due to the fact that they have overlapping interacting groups. In order to do so, we define a variant of our model in which the

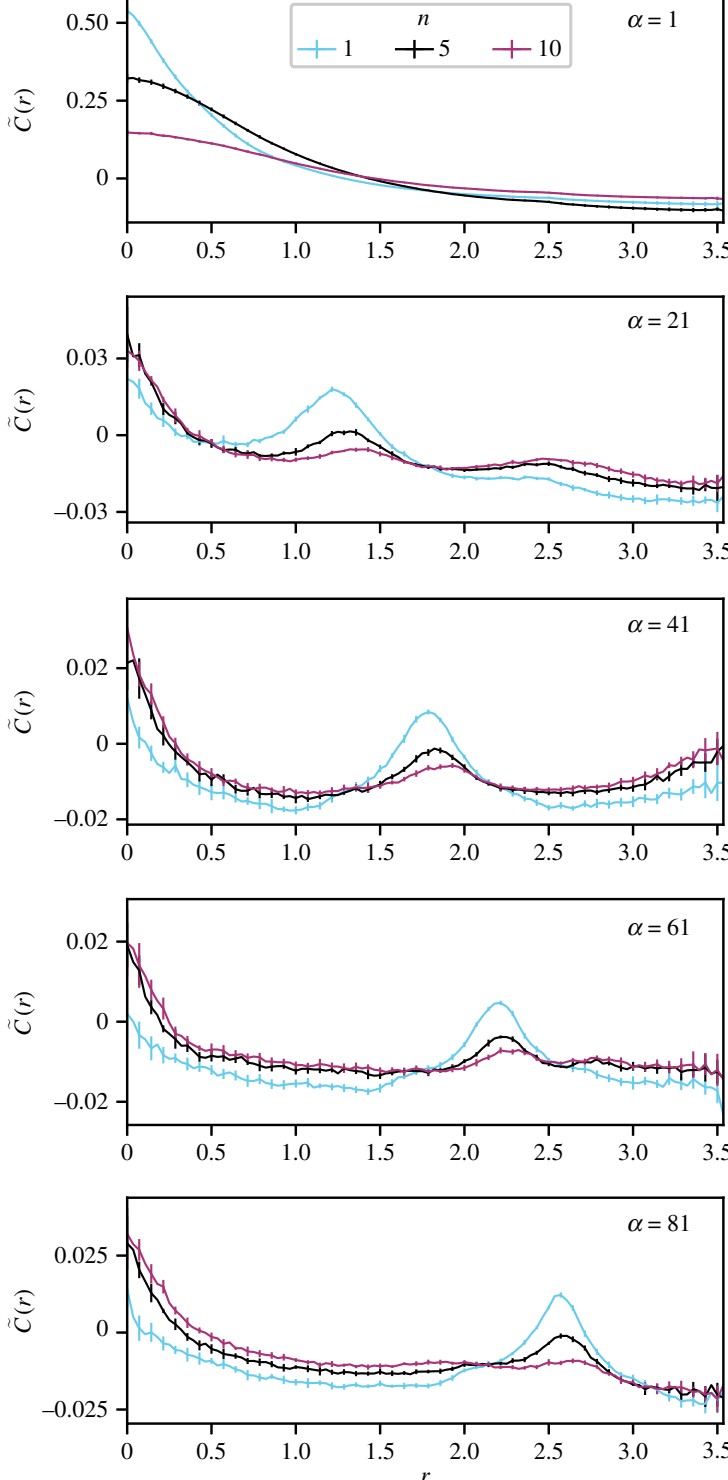

**Figure 2.** Velocity correlations at short distances emerge even from non-local interactions. For simulations of $N = 101$ agents, perhaps typical of a modest animal collective, we show the time-averaged pairwise correlation function $\tilde{C}(r)$ describing relative velocity correlation, in the centre of mass frame, as a function of metric distance $r$ between pairs of agents. Here the correlation function is reported for varying values of interaction range $\alpha$—the distance-order between an agent and those agents with which it explicitly co-aligns, termed its interacting group. Colours represent the number of agents in the interacting group: cyan $n = 1$, black $n = 5$ and purple $n = 10$. Noise is set to the crossover value of the order–disorder phase transition in each case (see electronic supplementary material, table S1 for noise values). In non-local regimes ($\alpha > 1$), a small secondary peak in the correlation function, arising due to the explicit alignment with the interacting group, is observed at a metric distance corresponding to members of the interacting group. In all regimes, however, a primary peak is observed at the shortest metric distances, persisting even when the interactions are explicitly non-local.

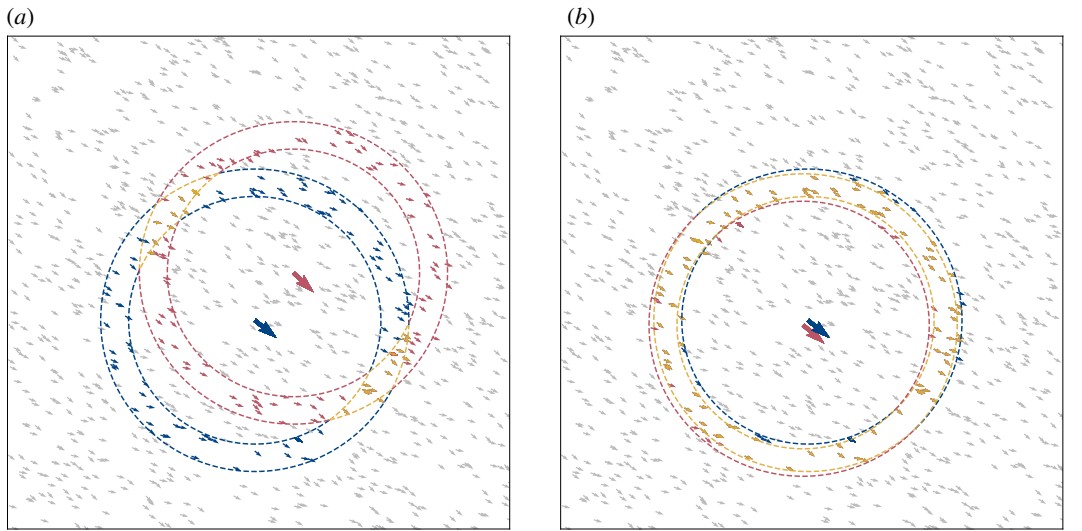

**Figure 3.** Schematic shows emergent three-body interactions between an arbitrary agent $i$ (blue and enlarged) and a neighbour $j$ (red and enlarged). The respective interacting groups (defined by $\alpha = 201$, $n = 100$) of $i$ and $j$ are coloured blue and red, while agents which are members of *both* interacting groups are shown in yellow. Dotted lines are shown as a guide to the eye to highlight the interacting groups. (*a*) Agent $i$ is the 50th nearest neighbour to $j$. (*b*) Here $i$ and $j$ are nearest neighbours. Nearby neighbours have higher overlap between their interacting groups. Even though primary agents $i$ and $j$ have no explicit interaction, the yellow agents influence both $i$ and $j$ and therefore provide an effective three-body interaction between them.

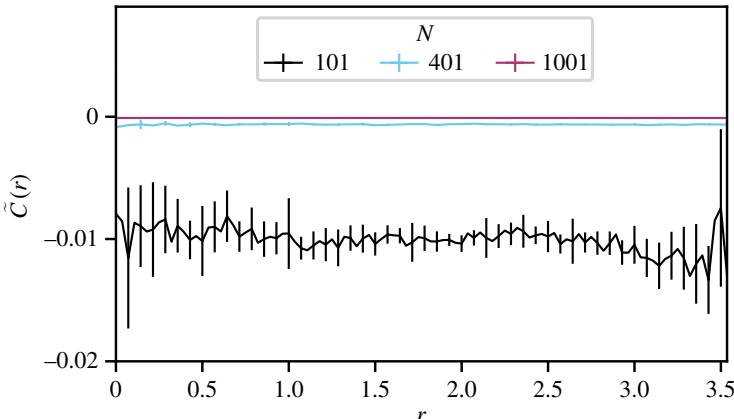

**Figure 4.** A model variant, that incorporates heterogeneous interaction ranges, generates no significant co-alignment in the centre of mass frame. Here we show the time-averaged velocity correlation function $C(r)$ as a function of metric distance $r$ between pairs of agents, colours represent different system size $N$ and number of agents in the interacting group $n$. Shown are results for systems with (i) black $N = 101$, $n = 10$ (ii) cyan $N = 401$, $n = 40$ and (iii) purple $N = 1001$, $n = 100$. The model variant is the same as the original model in all respects, except that the topological distance $\alpha_i$ from a given agent to the first member of its interacting group varies between agents. Previously, this was a constant value, now it is an integer uniform random variable (from the interval $(1, N - n)$) drawn once for each agent at the beginning of the simulation. Probably a consequence of the fact that agent's (positive) self-correlation with itself is omitted within our definition of the correlation function (equation (3.1)), for all systems we observe weak negative correlations at all ranges.

topological distance $\alpha_i$ from a given agent to the first member of its interacting group varies between agents. Previously, this was a constant value, shared by all agents. We run simulations in which the value of $\alpha_i$ is a uniform random integer (from the interval $(1, N - n)$) drawn once at the beginning of the simulation. All other aspects of the model remain unchanged.

The correlation function that we extract from these simulations is reported in figure 4. This is very different to the results of our previous simulations. The primary peak at short ranges is no longer observed, nor is the secondary peak. The fact that $\alpha_i$ varies between agents makes it unlikely that nearby neighbours have overlapping interacting groups. As such we do not observe the implicit three-body correlations.

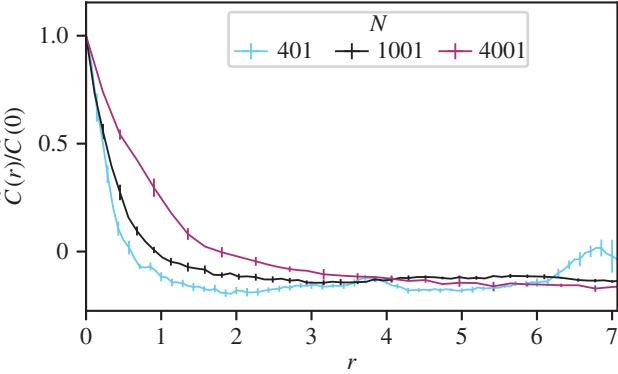

**Figure 5.** For all system sizes $N$, short-ranged correlations persist. Here we show the normalized time-averaged velocity correlation function $\tilde{C}(r)$ as a function of metric distance $r$ between pairs of agents, colours represent different system size $N$, interaction range $\alpha$ and number of agents in the interacting group $n$. Shown are results for systems with (i) cyan $N = 401$, $\alpha = 161$, $n = 40$ (ii) black $N = 1001$, $\alpha = 501$, $n = 100$ and (iii) purple $N = 4001$, $\alpha = 1601$, $n = 400$.

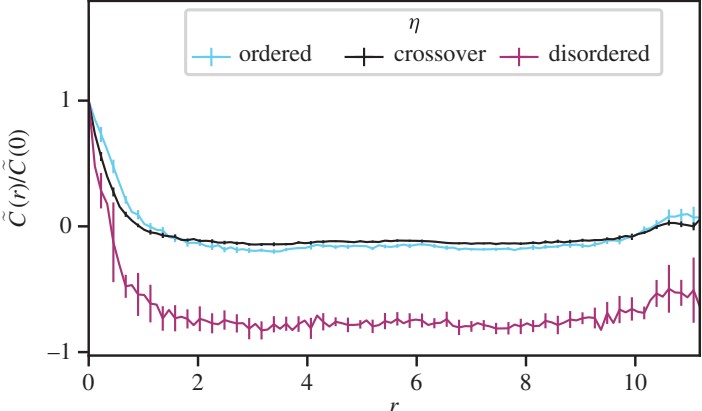

**Figure 6.** Above and below the crossover noise $\eta_c$ short-ranged correlations persist. Here we show the time-averaged velocity correlation function $C(r)$ as a function of metric distance $r$ between pairs of agents, for varying values of noise $\eta$. Shown are results for a system with $N = 1001$, $\alpha = 401$, $n = 100$. Colours represent different noise with (i) cyan, ordered ($\eta/\pi = 0.5$), (ii) black, crossover ($\eta/\pi = 1.76$) and (iii) purple, disordered ($\eta/\pi = 1.9$). The peak at short range in the disordered case is gradually lost on increasing noise or decreasing system size, see electronic supplementary material, figure S6.

This supports the interpretation provided of the main result of this paper: that the observed correlations in velocities of nearby agents can be explained by the overlapping of interacting groups, creating effective three-body interactions.

## 5. Conclusion

Our model gives rise to two forms of alignment: explicit alignment, in which agents align with those in their interacting group, and implicit alignment, in which nearby neighbours share members of their interacting group with which they both co-align, giving rise to interactions with a three-body character. We attribute nearest neighbour correlations to this implicit alignment. The concept of overlapping interacting groups is fundamental, even in collective motion models for which the interaction is short-ranged, e.g. [17,20]. In such models, each agent's interacting group consists of agents that are in close proximity and hence the implicit and explicit alignment effects are indistinguishable.

Our finding that the strongest velocity correlations occur between nearest neighbours is notable because experimental observations of velocity correlations in animal systems are used to infer which conspecifics are interacting with one another. The sensory range of many living self-propelled systems extends well beyond short range. Although it is widely assumed in the literature that local

interactions are responsible for the emergence of local correlations, and hence global order, we demonstrate that it is possible to obtain local velocity correlations even when the interactions are only between the most distant individuals. While our model is highly stylized we believe it reveals a more general principle for interactions between self-propelled particles that have long-ranged perception—nearby agents perceive the world in a similar manner and therefore can be expected to have corresponding behavioural similarities.

In conclusion, we have incorporated the range of co-alignment interactions as a tunable parameter within a model of collective motion. We demonstrate that nearest-neighbour velocity correlations emerge at all ranges of interaction—not only when the interactions are short-ranged. Furthermore, the strongest velocity correlations emerge at the shortest distances, even when the interactions are explicitly non-local.

Data accessibility. Source code for this research work is stored in GitHub: https://github.com/arthurebtking/Adjustable_Range_Model and has been archived within the Zenodo repository: https://doi.org/10.5281/zenodo.4429251. Dataset is archived within the Dryad Digital Repository: https://doi.org/10.5061/dryad.f4qrfj6vb [31].

Authors' contributions. A.E.B.T.K. performed all simulations and data analysis, including writing the source code. A.E.B.T.K. and M.S.T. conceived of the study and wrote the manuscript. All authors gave final approval for publication.

Competing interests. We declare we have no competing interests.

Funding. This study was supported by EPSRC (grant no. EP/L015374/1).

Acknowledgements. We acknowledge insightful comments made by the delegates at CECAM workshop: Emerging behaviour in active matter, Lincoln, 2019, where this work was first presented. A.E.B.T.K. also acknowledge funding from EPSRC under grant number EP/L015374/1, the Centre for Doctoral Training in Mathematics for Real-World Systems. Computational resources used for this research were provided by the Centre for Scientific Computing, University of Warwick.

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
