## [Peer Review File · Royal Society Open Science]

Review History

RSOS-201536.R0 (Original submission)

Review form: Reviewer 1

Is the manuscript scientifically sound in its present form?

Yes

Are the interpretations and conclusions justified by the results?

Yes

Is the language acceptable?

Yes

Do you have any ethical concerns with this paper?

No

Have you any concerns about statistical analyses in this paper?

No

Recommendation?

Accept with minor revision (please list in comments)

Comments to the Author(s)

I would like to thank the authors for reviewing and implementing most of the comments very carefully. I have few more comments concerning this new version.

The most important is that, although I appreciated the new experiments and figures being included as a response to my and other reviewer's comments, most of them have been included in the supplementary materials. This leaves the main text still very minimalist. I would like to encourage the authors to consider moving some of the supplementary materials results in the main manuscript, as this would make it much stronger (e.g. the counter example they have implemented as an answer to one of my comments, and also a discussion of the implications of this new case in the main text).

More minor comments:

The new introduction is better because it includes more discussion on the biological relevance, but somehow made the statement of the contribution (at the end of the introduction) shorter and slightly more indirect than before.

I do not understand why the term critical noise (which was very clear to me) has been replaced with crossover noise. I have read the answer to other reviewers but did not really understand the motivation. Is the crossover noise definition different from the one of the critical noise?

Review form: Reviewer 2

Is the manuscript scientifically sound in its present form?

No

Are the interpretations and conclusions justified by the results?

No

Is the language acceptable?

Yes

Do you have any ethical concerns with this paper?

No

Have you any concerns about statistical analyses in this paper?

No

Recommendation?

Major revision is needed (please make suggestions in comments)

Comments to the Author(s)

See file (Appendix A).

Decision letter (RSOS-201536.R0)

Dear Dr King

The Editors assigned to your paper RSOS-201536 "Non-local interactions in collective motion" have now received comments from reviewers and would like you to revise the paper in accordance with the reviewer comments and any comments from the Editors. Please note this decision does not guarantee eventual acceptance.

Please submit your revised manuscript and required files (see below) no later than 21 days from today's (ie 24-Nov-2020) date. Note: the ScholarOne system will 'lock' if submission of the revision is attempted 21 or more days after the deadline. If you do not think you will be able to meet this deadline please contact the editorial office immediately.

on behalf of Prof Miles Padgett (Subject Editor)
openscience@royalsociety.org

Associate Editor Comments to Author:

Please note that the reviewers have expressed a concern that the changes enacted after transfer from the Journal of the Royal Society Interface have not been sufficient. Please can you ensure that you not only fully respond to those earlier critiques and any new comments added in this round of review - if you do not do so, your paper may be rejected.

Reviewer comments to Author:

Reviewer: 1

Comments to the Author(s)

I would like to thank the authors for reviewing and implementing most of the comments very carefully. I have few more comments concerning this new version.

The most important is that, although I appreciated the new experiments and figures being included as a response to my and other reviewer's comments, most of them have been included in the supplementary materials. This leaves the main text still very minimalist. I would like to encourage the authors to consider moving some of the supplementary materials results in the main manuscript, as this would make it much stronger (e.g. the counter example they have implemented as an answer to one of my comments, and also a discussion of the implications of this new case in the main text).

More minor comments:

The new introduction is better because it includes more discussion on the biological relevance, but somehow made the statement of the contribution (at the end of the introduction) shorter and slightly more indirect than before.

I do not understand why the term critical noise (which was very clear to me) has been replaced with crossover noise. I have read the answer to other reviewers but did not really understand the motivation. Is the crossover noise definition different from the one of the critical noise?

Reviewer: 2

Comments to the Author(s)

See file

===PREPARING YOUR MANUSCRIPT===

===PREPARING YOUR REVISION IN SCHOLARONE===

Author's Response to Decision Letter for (RSOS-201536.R0)

See Appendix B.

RSOS-201536.R1 (Revision)

Review form: Reviewer 1

Is the manuscript scientifically sound in its present form?

Yes

Are the interpretations and conclusions justified by the results?

Yes

Is the language acceptable?

Yes

Do you have any ethical concerns with this paper?

No

Have you any concerns about statistical analyses in this paper?

No

Recommendation?

Accept as is

Comments to the Author(s)

Despite the bumpy review process for this paper (due to changing of the journal), the authors have addressed all my concerns.

I encourage the authors to run a final proof reading of their article, looking for small issues. For example, at line 24 of the introduction (last paragraph before section 2) there is a new sentence that starts with And.

Also, one of the produced proofs did not compile the references, I would make sure that the final proof does contain the references.

Review form: Reviewer 2

Is the manuscript scientifically sound in its present form?

Yes

Are the interpretations and conclusions justified by the results?

Yes

Is the language acceptable?

Yes

Do you have any ethical concerns with this paper?

No

Have you any concerns about statistical analyses in this paper?

No

Recommendation?

Accept as is

Comments to the Author(s)

I think paper is now ready for publication. The context of the work is much clearer. I'd like to thank the authors for implementing the necessary modifications.

Decision letter (RSOS-201536.R1)

Dear Dr King,

It is a pleasure to accept your manuscript entitled "Non-local interactions in collective motion" in its current form for publication in Royal Society Open Science. The comments of the reviewers who reviewed your manuscript are included at the foot of this letter.

You can expect to receive a proof of your article in the near future. Please contact the editorial office (openscience@royalsociety.org) and the production office (openscience_proofs@royalsociety.org) to let us know if you are likely to be away from e-mail contact – if you are going to be away, please nominate a co-author (if available) to manage the proofing process, and ensure they are copied into your email to the journal.

Please see the Royal Society Publishing guidance on how you may share your accepted author manuscript at <https://royalsociety.org/journals/ethics-policies/media-embargo/>. After publication, some additional ways to effectively promote your article can also be found here

<https://royalsociety.org/blog/2020/07/promoting-your-latest-paper-and-tracking-your-results/>.

on behalf of Professor Miles Padgett (Subject Editor)
openscience@royalsociety.org

Associate Editor Comments to Author:

Congratulations on your paper being accepted! Thank you for the support of the journal.

Reviewer comments to Author:

Reviewer: 2
Comments to the Author(s)

I think paper is now ready for publication. The context of the work is much clearer. I'd like to thank the authors for implementing the necessary modifications.

Reviewer: 1
Comments to the Author(s)

Despite the bumpy review process for this paper (due to changing of the journal), the authors have addressed all my concerns.

I encourage the authors to run a final proof reading of their article, looking for small issues. For example, at line 24 of the introduction (last paragraph before section 2) there is a new sentence that starts with And.

Also, one of the produced proofs did not compile the references, I would make sure that the final proof does contain the references.

Appendix A

Review for Non-local interactions in collective motion

The authors present a paper describing how a simple model of flocking can give rise to local velocity-velocity correlations from a model of arbitrary range. I have reviewed this paper twice previously and been critical of the way in which it presents its findings in the context of the existing literature, amongst other things. Indeed, it has taken me sometime to process this submission, partly because of work pressures, but mostly as I was attempting to understand how it varied from the last time I saw it. The responses to reviews from the first round are still present in the submission document. It appears the only change is that a "non-local" has been changed to "arbitrary range" in the abstract from what I can determine. All that being said, the paper is much improved in clarity, especially the methods, so I feel it is appropriate for publication, but the authors do need to remove some of the more unsubstantiated statements presenting previous literature.

I include some of my comments from as previous review, which from what I can determine have not been addressed

On lines 37-40 on page 3:

“We believe that it is fair to say that there is a consensus in the field that global order likely emerges as a result of local interactions between individuals [22, 23] and hence non-local interactions are often overlooked when considering model fitting of biological collectives. “

I contest this statement and don't believe it is fair to say it. The existence of references [25-31] contradicts this statement, for example. This observation would have been timely and novel in 2010, but not in 2020. My concern is that this ill-founded and slightly inflammatory sentence is a necessary part of the paper to justify novelty. I quote from two of the cited articles: “Local interactions with few neighbours are economic, and at the same time grant coherence at large scale. But why are interactions topological? What is the benefit of coordinating with the same number of individuals *independently* of their distance?” from Camperi et al. 2012 (ref 30) and “To date, it has not been clear of how these findings arise from the local interactions between individuals within the group.” Bode et al 2011 (ref 31) published in this journal 8 and 9 years ago, respectively.

At the end of the introduction L22-23 on page 4:

“Hence our work demonstrates the potential importance of non-local interactions to biological collectives.”

is unambiguously framed as the primary result, but this is neither novel, justified with real data on biological collectives nor is this the best demonstrated instance. Studies have concluded this, implicitly or explicitly, with reference to real data, for the last ten years.

With a more balanced presentation of previous work, the precise mechanism utilised in this article is suitable for publication, but I see this as only relevant to theoretical studies.

Appendix B

Response To Reviewers

Arthur E. B. T. King, Matthew S. Turner

8th December 2020

All new text we have added to the manuscript is highlighted in red.

Reviewer 1

I would like to thank the authors for reviewing and implementing most of the comments very carefully.

We appreciate the reviewer's comments.

I have few more comments concerning this new version. The most important is that, although I appreciated the new experiments and figures being included as a response to my and other reviewer's comments, most of them have been included in the supplementary materials. This leaves the main text still very minimalist. I would like to encourage the authors to consider moving some of the supplementary materials results in the main manuscript, as this would make it much stronger (e.g. the counter example they have implemented as an answer to one of my comments, and also a discussion of the implications of this new case in the main text).

We accept the referee's suggestion and have moved the section "Model variant: heterogeneous interacting groups" now titled "**Heterogeneity in the range between interacting groups**" into the main text, together with its associated Figure (now Fig 4).

More minor comments:

The new introduction is better because it includes more discussion on the biological relevance, but somehow made the statement of the contribution (at the end of the introduction) shorter and slightly more indirect than before.

We have made several changes to improve the introduction and also added a clarifying sentence in the last paragraph.

I do not understand why the term critical noise (which was very clear to me) has been replaced with crossover noise. I have read the answer to other reviewers but did not really understand the motivation. Is the crossover noise definition different from the one of the critical noise?

We are sensitive to strong feelings in the physics community about the use of the term “critical” in case this language is taken to suggest a deeper analogy with a critical point or that we have evidence to believe that the phase transition is continuous in nature. We reassure the referee that this is merely a cosmetic change in notation: As before, the crossover noise is determined from the inflection point of the order-disorder transition, in this sense it is identical to our previous “critical” noise.

Reviewer 2

The authors present a paper describing how a simple model of flocking can give rise to local velocity-velocity correlations from a model of arbitrary range. I have reviewed this paper twice previously and been critical of the way in which it presents its findings in the context of the existing literature, amongst other things. Indeed, it has taken me sometime to process this submission, partly because of work pressures, but mostly as I was attempting to understand how it varied from the last time I saw it. The responses to reviews from the first round are still present in the submission document. It appears the only change is that a “non-local” has been changed to “arbitrary range” in the abstract from what I can determine. All that being said, {the paper is much improved in clarity}, especially the methods, so I feel it is appropriate for publication, but the authors do need to remove some of the more unsubstantiated statements presenting previous literature.

We would like to thank Reviewer 2 for their comments and, in particular, the assessment that the clarity has improved. We reconsidered the comments made by reviewer 2 regarding “unsubstantiated statements” and have again modified our introduction. The statements we now make include,

- Numerical models indicate collective motion can arise from local interactions (unambiguously accurate)
- ...and it has been suggested that global order likely emerges as a result of local interactions between individuals (unambiguously accurate)
- this may explain why there has been a preference for focusing on local interactions when model fitting to biological collectives (in our view a reasonable deduction, qualified as speculative by the inclusion of the word “may”) .
- Non-local interactions are often overlooked when considering model fitting of biological collectives (correct on any reasonable statistical analysis of the literature).
- **We have now omitted the following:** We believe that it is fair to say that there is a consensus in the field that global order likely emerges as a result of local interactions between individuals.

I include some of my comments from a previous review, which from what I can determine have not been addressed

On lines 37-40 on page 3:

“We believe that it is fair to say that there is a consensus in the field that global order likely emerges as a result of local interactions between individuals [22, 23] and hence non-local interactions are often overlooked when considering model fitting of biological collectives.”

I contest this statement and don't believe it is fair to say it.

This statement has now been removed (even though we believe a survey of practitioners might still support our claim).

The existence of references [25-31] contradicts this statement, for example. This observation would have been timely and novel in 2010, but not in 2020. My concern is that this ill-founded and slightly inflammatory sentence is a necessary part of the paper to justify novelty. I quote from two of the cited articles: Local interactions with few neighbours are economic, and at the same time grant coherence at large scale. But why are interactions topological? What is the benefit of coordinating with the same number of individuals independently of their distance?from Camperi et al. 2012 (ref 30) and to date, it has not been clear of how these findings arise from the local interactions between individuals within the group. Bode et al 2011 (ref 31) published in this journal 8 and 9 years ago,

respectively.

We have removed the offending statement but, in case these criticisms cast a wider question on the significance of our work we would like to clarify the relevance some of the literature cited by the referee.

References [25-30] all include local interactions together with non-local interactions. It is therefore very difficult to know if any co-alignment is emerging due to the non-local interactions or not. In our assessment it is likely due to the local interactions in most, if not all, cases.

Reference [31], starts with the quote “Models have been highly influential in highlighting how collective motion can be produced from purely local interactions between individuals.” rather supporting our impression. In this paper Bode et al introduces a new model in which agents have a fixed spherical sensory zone, they may only interact with neighbours within this zone. Agents interact with a given neighbour with probability inversely proportional to distance. So the model allows non-local range interaction but is biased towards local interaction. The model also includes repel/align/attract interactions at short/medium/long ranges. The different ranges are defined with metric radii. This is not, in our view an unambiguously long-ranged model nor can general conclusions about long ranged interactions be drawn from it. The novelty of our work is examining non-local interaction in isolation via an interaction mechanisms with variable range, so the effects of local interaction can be directly compared with non-local interaction in the same framework.

At the end of the introduction L22-23 on page 4:

*“Hence our work demonstrates the potential importance of non-local interactions to biological collectives.”
is unambiguously framed as the primary result, but this is neither novel, justified with real data on biological collectives nor is this the best demonstrated instance. Studies have concluded this, implicitly or explicitly, with reference to real data, for the last ten years.*

We have changed the offending sentence to “Hence our work demonstrates the potential importance of non-local interactions **in generating alignment in** biological collectives.”

To be clear our primary result is **local correlations in velocity can arise purely from non-local interactions**. We are not aware of any publication that demonstrates this result.